Appendage abnormalities in spiders induced by an alternating temperature protocol in the context of recent advances in molecular spider embryology

Napiórkowska Teresa 1 tnapiork@umk.pl
Templin Julita 2
http://orcid.org/0000-0003-1987-9468 Napiórkowski Paweł 3
http://orcid.org/0000-0002-3834-7232 Townley Mark A. 4
1 Department of Invertebrate Zoology and Parasitology, Faculty of Biological and Veterinary Sciences, Nicolaus Copernicus University in Toruń , Toruń , Poland
2 Faculty of Biological and Veterinary Sciences, Department of Invertebrate Zoology and Parasitology, Nicolaus Copernicus University in Torun , Toruń , Poland
3 Department of Hydrobiology, Faculty of Biological Sciences, Kazimierz Wielki University in Bydgoszcz , Bydgoszcz , Poland
4 University Instrumentation Center, University of New Hampshire , Durham, New Hampshire , United States
Riesgo-Escovar Juan
Electronic publication date: 2023 Sep 7
Publication date: 2023
Volume: 11
Electronic Location ID: e16011
Received 2023 Jan 24; Accepted 2023 Aug 10
Copyright: © 2023 Napiórkowska et al.
Copyright year: 2023
Copyright holder: Napiórkowska et al.
License: This is an open access article distributed under the terms of the Creative Commons Attribution License, which permits unrestricted use, distribution, reproduction and adaptation in any medium and for any purpose provided that it is properly attributed. For attribution, the original author(s), title, publication source (PeerJ) and either DOI or URL of the article must be cited.
License URL: https://creativecommons.org/licenses/by/4.0/

Keywords: Developmental anomalies, Spider embryogenesis, Temperature fluctuations, Teratology, Thermally disturbed embryogenesis

Funding: Faculty of Biological and Veterinary Sciences of the Nicolaus Copernicus University Faculty of Biological Sciences of the Kazimierz Wielki University IDUB BENRISK This work was supported by the Faculty of Biological and Veterinary Sciences of the Nicolaus Copernicus University in Toruń (Poland) (statutory fund research) and by the Faculty of Biological Sciences of the Kazimierz Wielki University in Bydgoszcz (Poland). The research was funded by IDUB (BENRISK project). The funders had no role in study design, data collection and analysis, decision to publish, or preparation of the manuscript.

==============================
In the literature there are numerous reports of developmental deformities in arthropods collected in their natural habitat. Since such teratogenically affected individuals are found purely by chance, the causes of their defects are unknown. Numerous potential physical, mechanical, chemical, and biological teratogens have been considered and tested in the laboratory. Thermal shocks, frequently used in teratological research on the spider Eratigena atrica, have led to deformities on both the prosoma and the opisthosoma. In the 2020/2021 breeding season, by applying alternating temperatures (14 °C and 32 °C, changed every 12 h) for the first 10 days of embryonic development, we obtained 212 postembryos (out of 3,007) with the following anomalies: oligomely, heterosymely, bicephaly, schistomely, symely, polymely, complex anomalies, and others. From these we selected six spiders with defects on the prosoma and two with short appendages on the pedicel for further consideration. The latter cases seem particularly interesting because appendages do not normally develop on this body part, viewed as the first segment of the opisthosoma, and appear to represent examples of atavism. In view of the ongoing development of molecular techniques and recent research on developmental mechanisms in spiders, we believe the observed phenotypes may result, at least in part, from the erroneous suppression or expression of segmentation or appendage patterning genes. We consider “knockdown” experiments described in the literature as a means for generating hypotheses about the sources of temperature-induced body abnormalities in E. atrica.

Introduction

In natural aquatic and terrestrial habitats animals with body deformities are relatively common. This observation applies particularly to arthropods, including crustaceans, insects, myriapods, and chelicerates (e.g., Estrada-Peña, 2001; Asiain & Márquez, 2009; Leśniewska et al., 2009; Fernandez, Gregati & Bichuette, 2011; Feuillassier et al., 2012; Kozel & Novak, 2013; Scholtz, Ng & Moore, 2014; Di, Edgecombe & Sharma, 2018; Levesque et al., 2018; Brenneis & Scholtz, 2021). Since malformed arthropods are found purely by chance—e.g., during field research—the causes of their abnormalities remain unknown. Various hypotheses have attempted to explain the origin of these defects, sometimes affecting only one body part or organ, with a variety of physical, mechanical, chemical, and biological factors proposed (e.g., Miličić, Pavković-Lučić & Lučić, 2013).

Potential teratogenic factors can be tested in laboratory experiments using invertebrates, including species considered models for study (Lagadic & Caquet, 1998). A number of chemical reagents (e.g., Ehn, 1963a, 1963b; Itow & Sekiguchi, 1980; Köhler et al., 2005; Pinsino et al., 2010), radiation (Seitz, 1966, 1970; Yoshikura, 1969; Matranga et al., 2010), high humidity (Buczek, 2000), low/high temperature (Napiórkowska & Templin, 2012), and mechanical disturbance/manipulation (Holm, 1952; Sekiguchi, 1957; Scholtz & Brenneis, 2016) have already been exploited in teratology research. For instance, Holm (1940) hypothesized a teratogenic effect of temperature on spiders and later (e.g., Juberthie, 1968) investigated the effect of supraoptimal temperature on embryogenesis in harvestmen (Opiliones). Subsequently, the research was extended by using abrupt temperature changes during the incubation of Eratigena (formerly Tegenaria) atrica (C. L. Koch, 1843) embryos (e.g., Jacuński, 1984; Jacuński & Templin, 2003; Napiórkowska, Jacuński & Templin, 2010a, 2010b; Napiórkowska, Napiórkowski & Templin, 2016a, 2016b; Napiórkowska, Templin & Napiórkowski, 2021). It was observed that the application of alternating temperatures (lower and higher than the optimum) during early embryogenesis could lead to a range of deformities in both body tagmata. The most severe defects led to high embryo mortality or made it difficult for embryos to hatch to the postembryo stage. Moreover, some hatched but deformed individuals were unable to lead a normal life and achieve reproductive success. Anomalies described in E. atrica have included: oligomely (absence of one or more appendages), symely (fusion of contralateral appendages), schistomely (bifurcation of appendages), heterosymely (fusion of ipsilateral appendages), polymely (presence of one or more additional appendages), bicephaly (partial prosomal duplication), and so-called complex anomalies (two or more categories of anomaly occurring simultaneously) (e.g., Jacuński & Napiórkowska, 2000; Jacuński et al., 2002; Jacuński, Templin & Napiórkowska, 2005; Napiórkowska, Jacuński & Templin, 2007; Napiórkowska & Templin, 2012, 2013; Napiórkowska, Templin & Napiórkowski, 2013; Napiórkowska et al., 2016; Napiórkowska, Templin & Wołczuk, 2017).

At the Nicolaus Copernicus University in Toruń, Poland, teratological research on spiders by application of a thermal factor has been carried out since the 1970s (e.g., Mikulska, 1973). Since then, various anomalies have been described, with new cases recorded every year. Although early studies focused mainly on morphological description of teratologically altered individuals, attempts to explain the causes of deformities were also made. Jacuński (1984, 2002) suggested that many induced anomalies seen in late embryos or postembryos were presaged by structural aberrations evident in embryos as early as the blastoderm stage. For instance, thermal shocks led to the appearance of gaps in the blastoderm that Jacuński (1984) proposed may eliminate some embryo fragments, causing oligomely. On the other hand, it was suggested that symely or heterosymely could result if thermal treatment brought parts of the blastoderm closer together than was normal. However, Jacuński (2002) also noted that anomalies appearing well into embryonic development, e.g., during the limb bud elongation stage (see Wolff & Hilbrant, 2011), were not necessarily preceded by obvious structural abnormalities in earlier stages. This demonstrates that interpreting teratologies only in mechanistic terms is at best insufficient.

Over the last quarter-century, molecular techniques have been enlisted for the functional analysis of genes involved in the development of body segmentation and appendage formation in spiders. In particular, extensive use of in situ hybridization, RNA interference (RNAi), and immunolabeling have provided much insight into the expression of many developmental regulatory genes during spider embryogenesis (Damen & Tautz, 1998; Popadić et al., 1998; Abzhanov, Popadić & Kaufman, 1999; Schoppmeier & Damen, 2001; Stollewerk, Weller & Tautz, 2001; Stollewerk, Schoppmeier & Damen, 2003; Akiyama-Oda & Oda, 2003; Davis, D’Alessio & Patel, 2005; McGregor et al., 2008; Prpic, Schoppmeier & Damen, 2009; Schwager et al., 2009; Pechmann et al., 2011; Schwager, Meng & Extavour, 2015; Benton et al., 2016; Schwager et al., 2017; Oda & Akiyama-Oda, 2019; Heingård & Janssen, 2020; Setton & Sharma, 2021; to cite but a few such studies). This work has demonstrated various spider anomalies that result from the suppression or misexpression of specific segmentation and appendage patterning genes, raising the expectation that at least some abnormalities induced by thermal shock to embryos will be explicable in these terms. This does not preclude the possibility of mechanisms less directly related to gene expression, but also affected by temperature, also or alternatively being involved in creating defects. Thus, in this report, aberrant gene expression is broadly construed to include abnormal expression resulting from such temperature-influenced effects as atypical cell migration, cell division, cell death, and changes in metabolism. As such, overall expression of a gene could potentially be quantitatively normal but still present as abnormal phenotypes if positional or temporal perturbations to expression deviate substantially from normal.

In the 2020/2021 breeding season, using alternating temperatures during early embryogenesis of E. atrica, we obtained 212 postembryos with various body deformities. Since many of these anomalies have already been described in our previous works, we focused on those observed for the first time or those particularly relevant to evo-devo research, such as the rare cases where an appendage is found on the pedicel (petiolus, petiole) that connects the prosoma to the opisthosoma. Regarding the latter, appendages on one or both sides of the pedicel in postembryos of E. atrica were first described by Jacuński (1971, 1984). Some postembryos so afflicted did not survive beyond this stage, but for those that did, differences in the longevity of these appendages were later noted in Jacuński & Templin (1991): in one individual the appendage disappeared after the postembryo molted, while in another, a short, two-podomere appendage was present until the 6th stadium. Jacuński & Templin (1991) additionally described a postembryo of E. atrica with one substantial limb on the pedicel. Initially composed of four podomeres, by the 5th stadium it resembled, in form and segmentation, a complete walking leg, albeit distorted. During the 6th molt the leg broke off at the trochanter-femur joint. It grew back starting with the 7th molt, but the spider died during the 9th molt when loss of the leg re-occurred, accompanied by substantial loss of hemolymph. Jacuński & Templin (1991) proposed that the presence of an appendage on the pedicel is an atavistic trait. They also questioned whether the pedicel in spiders is correctly considered the first segment of the opisthosoma, since it has the potential to develop appendages similar in size and structure to walking legs.

Our study was aimed at further documenting the diversity of developmental anomalies that can be induced in E. atrica by applying the alternating temperature protocol to embryos. We also sought to consider abnormalities like those seen in the 2020/2021 breeding season in terms of potential errors in developmental gene expression. For the latter, we reviewed the literature related to the expression of such genes with a primary focus on functional studies that have employed RNAi to knock down specific genes in spiders.

Material and Methods

Teratological experiments on embryos of the spider Eratigena atrica (C. L. Koch, 1843) were carried out in the 2020/2021 breeding season. In September 2020, 32 sexually mature females and 24 males were collected from the vicinity of Toruń, Włocławek, and Chełmża, Poland. In the laboratory each individual was placed in a 250 cm3 well-ventilated glass container, kept in a darkened room. A temperature of 21 °C and a relative humidity (RH) of about 70% were maintained in the room throughout the experiment. Spiders were fed Tenebrio molitor larvae twice a week and water was supplied in soaked cotton balls. After 3 weeks, a male was introduced to each female for insemination. This procedure was repeated several days later with a different male to help ensure that all females were inseminated. First egg sacs were laid after a few weeks, followed periodically by additional egg sacs, averaging seven or eight egg sacs per female (in two previous breeding seasons) with up to 19 egg sacs constructed by a single female. All egg sacs were immediately removed from the containers and cut open to remove eggs, which were then counted and evenly divided into two groups: an experimental group and a control group. To verify that most eggs were fertilized, three randomly selected eggs per egg sac were immersed in paraffin oil and inspected.

Embryos from the control group were incubated at a temperature of 22 °C and 70% RH until hatching to the postembryo occurred, while embryos from the experimental group were exposed to alternating temperatures of 14 °C and 32 °C. The temperature was changed every 12 h for 10 days, until segments of the prosoma appeared on the germ band and limb buds appeared on these segments (comparable to Stage 9 in the trechaleid Cupiennius salei (Keyserling, 1877) (Wolff & Hilbrant, 2011); Stage 8.2 in the theridiid Parasteatoda (formerly Achaearanea) tepidariorum (C. L. Koch, 1841) (Mittmann & Wolff, 2012)). Subsequently, incubation was continued using the same conditions applied to the control group. After hatching, postembryos from both groups were examined for abnormalities in the prosoma and opisthosoma. Deformed individuals were photographed using a Zeiss Axiocam 105 color CMOS camera mounted on a Zeiss Axio Lab A1 light microscope and operated with Zen software (Version 2.3, blue edition).

We gathered references that present results of spider RNAi experiments, as they might shed light on potential gene misexpression leading to appendage abnormalities, as induced by the alternating temperature protocol.

Results

In the 2020/2021 breeding season, we obtained approximately 10,000 eggs/embryos, half of which constituted the control group. In this group, no hatched individuals with developmental defects were found, all postembryos having a properly developed prosoma with appendages and an opisthosoma with no observed abnormalities (Fig. 1). Approximately 10% of these controls failed to hatch, though development proceeded far enough in some that their fertilized status was apparent. The remainder, however, no doubt included some unfertilized eggs, with a comparable number presumably present in the experimental group, though we do not know what this number was. Eggs (100 or more) in the first egg sac built by a female are usually all fertilized or nearly so, but in subsequent egg sacs, which contain fewer total eggs, there are typically higher percentages of unfertilized eggs.

Figure 1 Eratigena atrica postembryo from control group, normally developed (ventral view).

Ch, chelicera; L1–L4, walking legs 1–4; OP, opisthosoma; PL, pedicel; Pp, pedipalp; PR, prosoma.

In the experimental group embryo mortality was much higher. About 40% of all embryos died at various stages of development; some failed to hatch from their eggshells even though their embryonic development appeared complete. In total, 3,007 postembryos were obtained in this group. Among these, individuals with a normally developed body structure predominated (2,795; 93%). The remaining postembryos (212) had various defects, most of which affected the prosoma and its appendages, although in nine individuals (4% of abnormal postembryos) deformities were also found in the opisthosoma. Oligomely was, by far, the most frequent anomaly, but multiple examples of each of several other types of anomaly—heterosymely, bicephaly, schistomely, symely, and polymely—were also obtained (Table 1). Moreover, >30% of postembryos displaying abnormal phenotypes did not fall neatly into one of these five types. They included individuals with complex anomalies, i.e., with multiple defects of more than one type, and those with abnormalities not conforming to any of these five types, grouped in Table 1 as ‘Other abnormalities’. The latter group included postembryos with significantly shortened or deformed appendages. Since many of the observed deformities have already been described in our previous studies, we present only selected cases, either recorded for the first time (Fig. 2) or of two postembryos with a short appendage on the pedicel (Fig. 3), constituting the only instances of polymely observed during this breeding season (Table 1).

Table 1 Types and frequency of anomalies in Eratigena atrica postembryos of the experimental group (i.e., subjected to the alternating temperature protocol).

No defects were observed in postembryos belonging to the control group.

Kind of anomaly	Number of individuals	%	
Oligomely	117	55.19	
Heterosymely	12	5.66	
Schistomely	6	2.83	
Bicephaly	7	3.30	
Symely	3	1.41	
Polymely	2	0.95	
Complex anomalies	28	13.21	
Other abnormalities	37	17.45	
Total	212	100.00	

Figure 2 Eratigena atrica postembryos with teratologic changes (ventral view).

(A) Postembryo lacking right chelicera and with a protuberance (‘a’) on the gnathocoxa of the right pedipalp; (B) postembryo with abnormally developed right chelicera (‘a’) and lacking right pedipalp; (C) postembryo lacking one of the right walking legs; (D) postembryo lacking right pedipalp and one of the left walking legs; (E) postembryo with schistomely of right second walking leg (L2), with its free ends of similar length labeled ‘a’ and ‘b’; (F) postembryo with deformed fourth walking leg (L4) on left side of the prosoma, with its shortened free ends labeled ‘a’ and ‘b’. Ch, chelicera; L1–L4, walking legs 1–4; Pp, pedipalp.

Figure 3 Ventral view of two Eratigena atrica postembryos with a short appendage on the pedicel (left A and B).

This appendage is enclosed by a white circle and shown enlarged (right A and B). Small protuberances on the pedicel appendage in (A) are labeled ‘a’ and ‘b’ (right). Contrary to the impression perhaps given by the image, these protuberances are not fused to L4. Ch, chelicera; L1–L4, walking legs 1–4; Pp, pedipalp.

The complex anomaly in the spider in Fig. 2A affected only the right side of the prosoma while the left side was formed normally with six well-developed, segmented appendages: chelicera, pedipalp, and four walking legs (L1–L4). On the right side of the prosoma the chelicera was missing (oligomely) and two appendages emerged from the gnathocoxa (= gnathendite = gnathobase = endite = maxilla), a normal pedipalp and a short protuberance (labeled ‘a’ in Fig. 2A) that lacked segmentation and moved independently. The legs were normally developed. The spider in Fig. 2B was likewise affected by a complex anomaly on the right side of the prosoma only. The chelicera was represented by a small, mobile protuberance (labeled ‘a’ in Fig. 2B) and the pedipalp was absent (oligomely). The legs had a normal structure. The spider in Fig. 2C was also affected by oligomely, the deformity most frequently observed in the teratological material. On the right side of the prosoma this individual had a well-developed chelicera and pedipalp, but only three legs. On the left side of the prosoma there was a complete set of appendages. Bilateral oligomely, though less common, was also observed in the teratological material. This anomaly affected the spider shown in Fig. 2D. On the right side of the prosoma there were five appendages—a chelicera and four legs—with the pedipalp missing. On the left side of the prosoma there were also only five appendages—a chelicera, pedipalp, and three legs; one leg was missing. The individual in Fig. 2E was affected by schistomely of leg L2 on the right side of the prosoma. The bifurcation started in the middle of the metatarsus and included the tarsus. The schistomely was symmetric in that the two distal ends (‘a’ and ‘b’ in Fig. 2E) were about the same length. The remaining appendages, including chelicerae and pedipalps, showed no irregularities. The spider in Fig. 2F had an especially unusual anomaly that affected leg L4 on the left side of the prosoma, presenting as a widened coxa from which only two short branches (‘a’ and ‘b’ in Fig. 2F) projected. The only visible segmentation on these branches was a single articulation, possibly demarcating the trochanter. This anomaly may represent schistomely initiated proximally within the developing leg, forestalling much further development. On the right side all appendages were well developed.

Figures 3A and 3B present a rare anomaly. These two postembryos had a very short appendage on the pedicel that connects the prosoma and opisthosoma. This additional appendage was on the left side of the pedicel. In both cases no other abnormalities were apparent. In the spider shown in Fig. 3A, the shortened appendage had the thickness of a walking leg, but it was not segmented. It had two small, rounded protrusions located prolaterally and distally (‘a’ and ‘b’ in Fig. 3A). In the spider in Fig. 3B, the appendage on the pedicel was of similar length and (proximally) width to that on the other specimen, and it was segmented to the extent that the first podomere (coxa) could be distinguished. The appendage widened distally, ending in an uneven surface with several bumps.

Discussion

Using the established thermal method for inducing developmental abnormalities in spider embryos (Jacuński, 1984), we obtained 212 individuals with body defects in the 2020/2021 breeding season, representing 7% (212/3,007) of the successfully hatched postembryos, and about 4% (212/5,000) of the embryos (hatched and unhatched), in the experimental group. These fairly low percentages suggest that spiders, as ectotherms, possess mechanisms that help make them relatively resistant to sudden temperature changes. One such mechanism likely includes the expression of heat shock protein (Hsp) genes, encoding protein-folding chaperones. It has been shown that the expression of Hsp genes significantly increases in response to various environmental stressors, including high temperature (Martínez-Paz et al., 2014 and references therein). Other mechanisms are presumably also involved as some induced morphological aberrations can be successfully eliminated by embryonic self-regulation and regeneration processes (Jacuński, 2002; Foelix, 2011; Oda et al., 2020; Oda & Akiyama-Oda, 2020; references therein). But the high mortality among experimental embryos (40%) as compared to control embryos (10%) also suggests a relatively high percentage of induced abnormality in the experimental group, severe enough to prevent hatching. It therefore appears that the alternating temperature protocol was effective in disrupting normal development in about one-third of embryos, causing a range of developmental anomalies and high embryo mortality. This thesis is supported by an absence of developmental defects and low embryo mortality within the control group.

We have noted a trend for mortality percentages in both control and experimental groups to rise over the past decade (Napiórkowska, Templin & Napiórkowski, 2013; Napiórkowska, Napiórkowski & Templin, 2016a; Napiórkowska et al., 2016; Napiórkowska & Templin, 2017; Napiórkowska, Templin & Napiórkowski, 2021), from a low of 4% and 20%, respectively (Napiórkowska, Templin & Napiórkowski, 2013), to the present study’s high (10%, 40%, respectively). Conversely, the percentage of successfully hatched postembryos in the experimental group that exhibited defects has shown a downward trend over the same period, from highs of 17–18% (Napiórkowska, Templin & Napiórkowski, 2013; Napiórkowska et al., 2016) to a low of about 4% (Napiórkowska, Templin & Napiórkowski, 2021), rebounding moderately in the present study with 7%. These opposite trends in the experimental group could be related: if a larger percentage of embryos adversely affected by the alternating temperature protocol fail to hatch, a smaller percentage of defective individuals may remain among the embryos that hatch successfully. As yet we have no explanations for these trends. We have not knowingly made any changes to our procedures in this period. Field-collected adults have been captured, and control group spiderlings released, in the same locations throughout this period. Conceivably, our collection and release activities at these sites may be generating or contributing to the trends. This can be investigated by comparing, during the same breeding season, mortality and defect percentages between the progeny of adults obtained from our usual collection/release sites with progeny of adults collected from distant virgin sites. Other factors potentially contributing to the observed trends, such as the influence of climatic changes on reproduction in E. atrica, may also be profitably investigated.

In the teratological material, oligomely was the most frequent anomaly by a large margin, accounting for about 55% of cases, and it was even more prevalent considering that oligomely was a component in some postembryos (e.g., Figs. 2A and 2B) categorized as having ‘Complex anomalies’ (Table 1). Other anomaly categories were observed much less frequently, which agrees with the results of previous studies. If we express percentages by considering only the six conspicuous single anomaly categories (i.e., discounting ‘Complex anomalies’ and ‘Other abnormalities’ categories) as they occurred on prosomata and pedicels, cases of oligomely accounted for 79.6% of defects in this study. This percentage, across five earlier studies (Jacuński, 1984; Napiórkowska, Templin & Napiórkowski, 2013; Napiórkowska et al., 2016; Napiórkowska & Templin, 2017; Napiórkowska, Templin & Napiórkowski, 2021), ranged from 73.5–84.8%. In contrast, percentages for the other five single anomaly categories were (given as % for this study followed by % range in the five earlier studies): heterosymely, 8.2%, 4.9–10.4%; schistomely, 4.1%, 2.2–9.9%; bicephaly, 4.8%, 0–6.5%; symely, 2.0%, 0–7.8%; polymely, 1.4%, 0–3.7%.

It remains to be determined why instances of oligomely dominate among teratological postembryos that have been subjected to the alternating temperature protocol as embryos. One important consideration is that percentages of different anomaly types presented in this and earlier studies reflect their occurrence in successfully hatched postembryos. Thus, the first step in addressing the question of oligomely prevalence is to determine if these percentages agree with percentages of defect types as they exist in the embryo stage. It is possible that oligomelic embryos are more likely to survive and successfully hatch than embryos exhibiting other defect types and consequently oligomely is better represented among postembryos than among embryos. We therefore intend to explore the feasibility of ascertaining anomaly types on a large scale in late embryos.

Oligomelic postembryos

Molecular embryological research has suggested alterations to normal gene expression that might account for some instances of appendage loss. Parental RNAi (pRNAi) studies, especially in P. tepidariorum, have revealed a range of abnormal phenotypes from knockdown of selected developmental genes (Oda & Akiyama-Oda, 2020), depending on the specific gene suppressed and on the degree of suppression of a given gene within different embryos. These phenotypes can include embryos exhibiting oligomely, though in some instances lethal abnormalities co-occur, indicating that widespread down-regulation of the targeted genes does not account for oligomelic postembryos like those in Figs. 2A–2D. For example, knockdown of the Notch-signaling-pathway component Delta in P. tepidariorum (Pt-Delta) (Oda et al., 2007) or the spider gap gene Pt-Sox21b.1 (Paese et al., 2018; Baudouin-Gonzalez et al., 2021) results in loss of leg-bearing segments, but this is accompanied by loss of all opisthosomal segments. More localized suppression, however, comparable to that achieved by embryonic RNAi (eRNAi) (Oda & Akiyama-Oda, 2020), cannot be ruled out in oligomelic postembryos.

As an aside, conspicuously lethal consequences of gene downregulation, as occur with knockdown of genes such as Pt-Delta and Pt-Sox21b.1, are potentially relevant to the high mortality that was observed in experimental E. atrica embryos. Equally lethal, though less conspicuous, is embryonic development that, superficially, proceeds essentially to completion without obvious defect, but the embryo nevertheless fails to hatch. Embryos like these were among the 40% of the experimental group that did not hatch. It is thus worth noting that fully developed embryos, not exhibiting defects but unable to hatch, were produced with high frequency when three transcription factors, Pt-foxQ2, Pt-six3.1, or Pt-six3.2, were individually suppressed by pRNAi (Schacht, Schomburg & Bucher, 2020). We hasten to emphasize, however, that there may well be many mechanisms by which this inability to hatch is produced, with lethal consequence, including some unrelated to abnormal gene expression. Here and elsewhere in this discussion, in noting similarities between phenotypes obtained by thermal shock and by RNAi of specific genes, the involvement of those genes in producing the thermally-induced abnormalities, while a possibility, is by no means assured and certainly no such definitive claim is intended.

More likely to be involved in appendage losses like those in Fig. 2 are genes that, when knocked down, result in oligomelic embryos able to survive hatching. Examples of two such genes, expressed during early embryogenesis within the period our thermal treatment is applied, are the gap gene hunchback (hb) (Schwager et al., 2009) and Distal-less (Dll), an appendage patterning gene that also plays an earlier gap gene role in spiders (Pechmann et al., 2011). pRNAi of hb in P. tepidariorum (Pt-hb) yielded postembryos missing the L2 leg pair or both L1 and L2 legs (Schwager et al., 2009), while, similarly, pRNAi of Pt-Dll produced postembryos lacking the L1 leg pair or both L1 and L2 legs (Pechmann et al., 2011; Setton et al., 2017; Setton & Sharma, 2018). These losses reflected loss of the segments on which the legs would have developed and it was only segments bearing walking legs that were so affected (Schwager et al., 2009; Pechmann et al., 2011), reflecting the distinction between segmentation of the head region, with its chelicerae and pedipalps, and that of the thorax region, with its four pairs of legs (Kanayama et al., 2011). If this also applies to E. atrica, then abnormal suppression of Ea-hb would not contribute to oligomely involving chelicerae and pedipalps (Figs. 2A, 2B, and 2D), but it could be a factor in spiders with missing legs (Figs. 2C and 2D). The same can be said for Ea-Dll suppression during its early involvement with prosomal segmentation (its gap gene role) (Pechmann et al., 2011). Later suppression of Dll in limb buds (Chen, Piel & Monteiro, 2016), whether preceded by early Dll suppression (pRNAi; Pechmann et al., 2011) or not (eRNAi; Schoppmeier & Damen, 2001; Pechmann et al., 2011), resulted in truncated appendages but not in any additional appendage loss.

There are, however, two confounding considerations where potential abnormal Ea-hb or Ea-Dll expression is concerned: (1) Though Schwager et al. (2009) did note left-right leg reduction asymmetry in Pt-hb pRNAi embryos, leg losses resulting from prosomal segment losses have usually been symmetric (Schwager et al., 2009; Pechmann et al., 2011), whereas the alternating temperature treatment applied in this study has often yielded asymmetric (Figs. 2C and 2D), as well as symmetric (e.g., Jacuński, Templin & Napiórkowska, 2005), leg oligomely. (2) We have not been able to determine which legs specifically have been missing in oligomelic postembryos, even after examining leg neuromeres in histological sections (Jacuński, Templin & Napiórkowska, 2005; Napiórkowska, Napiórkowski & Templin, 2016b), and therefore we do not know if leg losses have been consistent with Ea-hb or early Ea-Dll suppression.

Regarding (1), if, speculatively, Ea-hb or Ea-Dll is inhibited by our alternating temperature protocol (at this point possibilities only), such inhibition might be more localized, random, and asymmetric than that often resulting from pRNAi. Indeed, unilaterally oligomelic E. atrica with corresponding unilateral losses of leg nerves and ganglia indicate that thermally-induced disturbances result in losses of hemisegments more often than of full segments (Jacuński, 1983; Jacuński, Templin & Napiórkowska, 2005; Napiórkowska, Napiórkowski & Templin, 2016b). This is reminiscent of asymmetric prosomal appendage shortening that has been induced in C. salei by knockdown of Cs-Dll using eRNAi (Schoppmeier & Damen, 2001) and of seven-legged postembryos that have occasionally resulted from Pt-Dll pRNAi, indicating loss of a single L1 hemisegment (Setton et al., 2017). Schoppmeier & Damen (2001) noted the median furrow (ventral sulcus) that divides the right and left halves of the embryonic germ band (Foelix, 2011; Wolff & Hilbrant, 2011), and the seemingly independent development of the two halves, as a possible explanation for such asymmetric phenotypes.

Regarding (2), future studies could explore a strategy used by Pechmann et al. (2011) for ascertaining the identity of missing legs: for oligomelic postembryos able to molt successfully to at least 1st instars, the number and arrangement of slit sense organs on the sternum, compared to control spiders, should help identify the missing legs and provide an alternative to histological sectioning for indicating if symmetric/asymmetric oligomely of legs is accompanied by loss of an entire segment/hemisegment, as previously suggested based on histology (Jacuński, 1983; Jacuński, Templin & Napiórkowska, 2005; Napiórkowska, Napiórkowski & Templin, 2015, 2016b).

We should also note that, unlike RNAi experiments, in which the gene targeted by treatment is known, genes most directly impacted by application of the alternating temperature protocol may be cofactors, upstream regulators, or downstream targets of genes discussed here as being potentially perturbed by the protocol, rather than directly affecting expression of the candidate gene itself. For example, pRNAi of the transcription factor Sp6-9 in P. tepidariorum (Pt-Sp6-9) has been observed to reduce or eliminate Pt-Dll expression (Königsmann et al., 2017; Setton & Sharma, 2018) as well as eliminate expression of the segment polarity gene Pt-engrailed-1 (Pt-en-1) in the L1 and L2 segments (Setton & Sharma, 2018), similar to the effect of Pt-Dll pRNAi on Pt-en-1 expression (Pechmann et al., 2011). Resulting phenotypes included embryos missing these two segments and, so, also the legs that would form on them (Königsmann et al., 2017; Setton & Sharma, 2018). Thus, in this example, defects consistent with inhibited Ea-Dll expression could, hypothetically, arise via thermally-induced direct disruption to a different member of the same gene network, namely Ea-Sp6-9 (Königsmann et al., 2017; Setton & Sharma, 2018). Also, genes most directly affected may vary among embryos depending on, e.g., the exact timing of a temperature switch in relation to an embryo’s stage of development. It is also worth repeating that thermally-induced perturbations to normal gene expression might have abnormal spatial or temporal components in addition to, or rather than, quantitative aberrations.

On first consideration, missing pedipalps, as in Figs. 2B and 2D, could suggest disturbance to the normal expression of the Hox gene labial (lab), specifically the paralog lab-1 (lab-B in Schwager et al., 2017), first expressed at Stage 4 in P. tepidariorum (Pechmann et al., 2015). Its knockdown by pRNAi can result in postembryos lacking pedipalps, though, unlike leg losses that are due to loss of the corresponding prosomal segments, the pedipalpal segment is retained (Pechmann et al., 2015). On the other hand, like the above pRNAi-induced leg losses, pedipalp loss as seen in Pt-lab-1 pRNAi postembryos has been symmetric (Pechmann et al., 2015), whereas the alternating temperature treatment more often results in asymmetric pedipalp oligomely in E. atrica (Figs. 2B and 2D), suggesting a potential localized disruption to Ea-lab-1 expression. However, an abnormal postembryo like that shown in Fig. 2B, in which the site of a missing pedipalp is adjacent to a greatly reduced chelicera (labeled ‘a’), does not support this suggestion if we assume a shared genetic cause for both anomalies (this assumption is by no means certain). This is because expression of lab-1 (or any of the Hox genes) is not involved in specifying chelicera morphology (Pechmann et al., 2010).

An alternative explanation that might encompass both defects has not yet emerged from functional studies in spiders. The gene dachshund-2 is expressed proximally in both chelicerae and pedipalps, but the only noted phenotypic consequences of its knockdown by pRNAi in P. tepidariorum are malformed patellae in the walking legs (Turetzek et al., 2015). Two paralogs of extradenticle (exd-1, exd-2) and homothorax-1 (hth-1) are also expressed proximally in pedipalps and chelicerae (Prpic & Damen, 2004; Pechmann & Prpic, 2009; Turetzek et al., 2017), but exd has not been the subject of RNAi experiments in spiders, or any chelicerates (Nolan, Santibáñez-López & Sharma, 2020), and among chelicerates hth function has only been examined by eRNAi in the harvestman Phalangium opilio Linnaeus, 1758 (Sharma et al., 2015). However, studies in insects and spiders indicate that exd-1 and hth-1 of spiders are functionally linked (Hth-1 required for translocation of Exd-1 into the nucleus), such that knockdown of either gene would likely produce similar, though not identical, phenotypes (Sharma et al., 2015; Turetzek et al., 2017; references therein). Phenotypes resulting from knockdown of the single-copy hth in P. opilio (Po-hth) included homeotic transformations of chelicerae and pedipalps to leg identities, appendage truncation, and fusions between chelicerae and pedipalps, though, importantly, apparently not pedipalp oligomely (the results do, however, state “The labrum and/or some appendages also failed to form” (among Class I phenotype embryos) (Sharma et al., 2015) without elaboration). Interestingly, like the aforementioned defect asymmetry observed in Cs-Dll eRNAi C. salei embryos (Schoppmeier & Damen, 2001), a high incidence of asymmetric defects was also obtained with Po-hth eRNAi P. opilio embryos (Sharma et al., 2015). As mentioned, asymmetric defects are likewise often obtained by the alternating temperature protocol. These three examples demonstrate that aberrations on one side of the germ band do not necessarily affect the other side (Schoppmeier & Damen, 2001) and suggest limited, non-global perturbations to gene expression or other developmental processes.

Postembryos with schistomely or in ‘Other abnormalities’ category

Appendage development relies on differentiation along proximal-distal (P-D), dorsal-ventral (D-V), and anterior-posterior (A-P) axes, the last especially little studied in spiders. Genes involved with establishing these axes may be susceptible to thermally-induced abnormal expression, resulting in limb malformations. For example, a key player in establishing the D-V axis is the gene FoxB, encoding a forkhead box transcription factor that is ventrally expressed within appendages (Heingård, 2017; Heingård et al., 2019). Its knockdown in P. tepidariorum by pRNAi resulted in greatly reduced hatching success and altered expression of downstream genes that normally show ventral (wingless (Pt-wg/Wnt1), Pt-H15-2), dorsal (optomotor-blind (Pt-omb)), and distal (decapentaplegic (Pt-dpp)) expression within appendages, resulting in ‘dorsalized’ legs and pedipalps (Heingård, 2017; Heingård et al., 2019). Such Pt-FoxB pRNAi embryos that were able to hatch successfully and progress to the 1st stadium exhibited distally crooked legs and pedipalps, comparable to some postembryos included in our ‘Other abnormalities’ category (Table 1). This category also included postembryos with significantly shortened appendages, a phenotype that has also been observed in mildly affected Pt-Sp6-9 pRNAi embryos and postembryos, and has included asymmetric defects (Königsmann et al., 2017; Setton & Sharma, 2018).

Appendage bifurcation, i.e., schistomely (Figs. 2E and 2F), in postembryos might also be considered in terms of erroneous expression of genes modeling the appendage axes, with schistomely representing distal duplication of the P-D axis (Cotoras, de Castanheira & Sharma, 2021). Though functional data (e.g., RNAi) are lacking in chelicerates (Cotoras, de Castanheira & Sharma, 2021), expression data in P. tepidariorum for dpp (Akiyama-Oda & Oda, 2003) and wg/Wnt1 (Janssen et al., 2010), among other evidence from spiders and other arthropods (Pechmann et al., 2010), have been consistent with dpp and wg/Wnt1 expression early in spider appendage development initiating a gene cascade that generates the P-D axis (Prpic et al., 2003). In legs and pedipalps, three distinct domains of expression establish the P-D axis via expression of Dll distally, dachshund-1 (dac-1) medially, and exd-1/hth-1 proximally (Prpic & Damen, 2004; Pechmann et al., 2010). Disturbances in the normal expression of dpp, wg/Wnt1, or their downstream targets caused by thermal shocks may result in a duplication of the P-D axis. In a report of cheliceral schistomely in the spider Tetragnatha versicolor Walckenaer, 1841, Cotoras, de Castanheira & Sharma (2021) hypothesized that the defect could be replicated by introducing ectopic Dpp and Wg/Wnt1. The schistomely shown in Fig. 2E, at the distal end of a leg, suggests perturbations that included direct or indirect abnormality in Dll expression while the more proximal schistomely indicated in Fig. 2F, on a noticeably wider appendage than the normal legs, potentially represents abnormal expression of dpp, wg/Wnt1, and dac-1 (among other possibilities), the latter’s expression coincident with the trochanter and femur (Abzhanov & Kaufman, 2000; Prpic et al., 2003; Prpic & Damen, 2004).

Postembryos exhibiting pedicel polymely

Arguably the most interesting cases from the perspective of evolutionary/developmental biology involve two individuals with an appendage on the pedicel (first segment of the opisthosoma, O1; in spiders, coincident with somite VII) that are presented in Figs. 3A and 3B. Appendages do not usually form on the O1 segment in spiders and such defects are rare even among E. atrica subjected to alternating temperatures as embryos. Within this segment, the principal Hox genes expressed are the two paralogs of Antennapedia (Antp) (Damen et al., 1998; Khadjeh et al., 2012; Schwager et al., 2017). Knockdown of Antp-1 in P. tepidariorum (Pt-Antp-1) by pRNAi has demonstrated that it is responsible for repressing the development of legs on the O1 segment (Khadjeh et al., 2012). At its most severe, this down-regulation of Pt-Antp-1 resulted in sufficient de-repression of leg development in O1 that 10 walking legs formed; the usual eight plus a pair on the pedicel that were like the former morphologically and in lateral placement except a little shorter and thinner (Khadjeh et al., 2012; replicated by Setton & Sharma, 2018). Expression of the genes that establish the P-D axis in legs (Pt-exd-1, Pt-hth-1, Pt-dac-1, Pt-Dll) was nearly identical between the ectopic O1 legs and normal L1–L4 legs. Moreover, expression of the Hox genes Deformed-A (Pt-Dfd-A) and Sex combs reduced-B (Pt-Scr-B; paralogs as designated in Schwager et al., 2017) within the 10 legs indicated that the ectopic legs on O1 were not homeotic copies of any of the normal walking legs, but they were instead true O1 segment de-repressed legs (Khadjeh et al., 2012).

It is of interest that Khadjeh et al. (2012) obtained not only severely affected postembryos with a pair of complete legs on the pedicel following knockdown of Pt-Antp-1, but in more moderately affected individuals they observed only short leg-like projections on the pedicel. Further, in a triple pRNAi experiment (to suppress Pt-Antp-1 and two other Hox genes), they obtained two postembryos with an incomplete appendage on just one side of the pedicel. They attributed this asymmetric (“mosaic”) phenotype to the lesser quantity of each dsRNA that could be injected when attempting to inhibit three genes simultaneously, resulting in less effective suppression of Pt-Antp-1. This range of outcomes is again reminiscent of the results obtained when alternating temperatures are applied to embryos of E. atrica, where appendages may form on the pedicel symmetrically or only on one side (Fig. 3), and these appendages may exhibit little or considerable development, from a short, unsegmented projection to a segmented, essentially complete leg (Jacuński, 1971, 1984; Jacuński & Templin, 1991; this study). This suggests that the alternating temperature protocol has the potential to disturb, to varying extent, normal expression of Ea-Antp-1 or associated up- or downstream genes in the O1 segment.

There is a long history of embryological observations on spiders that indicates an ancestry in which appendages were present on somite VII (e.g., Korschelt & Heider, 1890; Jaworowski, 1896; Janeck, 1909; Yoshikura, 1954; Yoshikura, 1955; Wolff & Hilbrant, 2011). Principally, this is indicated by a small, short-lived protuberance or patch, sometimes explicitly interpreted as an incipient limb bud, appearing on each O1 hemisegment when the opisthosomal limb buds develop. These transient O1 limb buds apparently do not form in all spider taxa (Dawydoff, 1949), however, as they have not been noted in some detailed embryological studies (Montgomery, 1909; Holm, 1940; Rempel, 1957; Mittmann & Wolff, 2012; Pechmann, 2020). It is notable that putative limb buds on O1 have been observed in Heptathela (Yoshikura, 1954; Yoshikura, 1955), a member of the basal Mesothelae, as well as in several members of the derived araneomorph RTA clade, to which E. atrica belongs (Wheeler et al., 2017).

Considering that small, transitory protrusions (potential appendages) may appear on the pedicel (O1) segment in embryonic spiders, and that by use of targeted gene suppression (pRNAi) it is possible to obtain appendages on the pedicel with the structure of walking legs that nevertheless have their own O1 identity (Khadjeh et al., 2012), it might be worth reconsidering whether somite VII, the pedicel, is indeed the first segment of the opisthosoma, as it is usually described, rather than the last segment of the prosoma. This thought is stimulated by another result obtained by Khadjeh et al. (2012); that limb repression also occurs as a normal part of development in the O2 segment (somite VIII), but when the genes that redundantly promote this repression (Pt-Antp-1, Ultrabithorax-1 (Pt-Ubx-1)) are suppressed by double pRNAi, the ectopic appendages that form on O2 appear far more vestigial than the legs induced to form on O1. This may reflect less effective overall de-repression in O2 because of the repression redundancy present in O2, not shared by O1, but it could also conceivably reflect an early euchelicerate ancestry in which appendages on somites VII and VIII differed substantially in morphology, with those on VII more limb-like and those on VIII more plate-like, suggestive of a border between tagmata. Such a difference in appendage morphology has been interpreted for the Devonian euchelicerate Weinbergina and is also seen in extant Xiphosurida (horseshoe crabs) (Dunlop & Lamsdell, 2017).

Applying Lamsdell’s (2013:4) definition of a tagma as “…a distinct and discrete morphological region that comprises a series of equivalently modified appendages that constitute a unit of specific form…or sometimes function…”, the traditional view of the O1 segment as part of the spider opisthosoma seems appropriate. Both the normally legless condition of the pedicel and the maneuverability it imparts to the rest of the opisthosoma (Dunlop & Lamsdell, 2017) suggest a form and function more in keeping with those of the opisthosoma. In addition, during spider embryogenesis, the germ band initially divides into the prosomal segments and a posterior ‘segment addition zone’ (SAZ) from which the opisthosomal segments, including O1, subsequently derive in anterior-to-posterior sequence (Schwager et al., 2015). These differing paths to segmentation in the two tagmata also favor an opisthosomal identity for the O1 segment.

On the other hand, Lamsdell (2013) and Dunlop & Lamsdell (2017) acknowledge that establishing borders between tagmata can be difficult because the ends of a tagma and their associated appendages may differ substantially from the rest of the tagma. The border between prosoma and opisthosoma, with somite VII’s questionable affiliation, is given as a prime problematic example (Dunlop & Lamsdell, 2017). They review evidence from fossil and extant chelicerates that supports a chelicerate groundplan in which somite VII is prosomal, as suggested by Stürmer & Bergström (1981). This possibility is further supported by the potential for appendages with leg-like morphology to develop on the spider pedicel, whether induced by application of pRNAi or alternating temperatures, and, along with transitory limb bud formation on the O1 segment in some spiders, suggests loss of somite VII appendages present in basal euchelicerate ancestors of arachnids (Dunlop & Lamsdell, 2017). Thus, an interpretation of atavism for appendages developing on the pedicel in teratological spiders (Jacuński, 1971, 1984; Jacuński & Templin, 1991) remains valid. Also noteworthy is the observation that, in some chelicerates, walking leg segments (all or just L4), as well as the opisthosomal segments, are derived from the SAZ and, in one known instance (a mite), O1 segmentation precedes that of L4 (reviewed in Schwager et al., 2015). Thus, it seems the mechanism of segmentation during embryonic development does not necessarily provide a reliable means for assigning segments to tagmata in a way that agrees with morphological/functional regions.

Summary and future directions

By applying alternating temperatures during early spider embryogenesis, we obtained high embryo mortality, changes in number, size, and shape of appendages or their podomeres, and formation of appendages on the pedicel; a body segment (O1 = somite VII) on which appendages are not normally found in spiders. Thus, by using appropriate methods, abnormalities can be induced that potentially reflect certain ancestral traits present in basal (eu)chelicerates, including possibly atavistic appendages on segment O1. This type of developmental abnormality has a bearing on the question of the tagma to which somite VII belongs, prosoma or opisthosoma, with implications tied to chelicerate phylogeny.

Based on recent research on genes that determine the formation of segments and appendages, we suspect that at least some of the observed developmental defects arising from our alternating temperature protocol are the result of blocked or otherwise aberrant expression of relevant genes, including Hox genes. Atypical expression may potentially include spatial and temporal, as well as quantitative, deviations from normal. Though the possible involvement of specific genes as discussed above is speculative, it is one step toward the goal of testing hypotheses that attribute specific anomaly types to disturbances affecting specific genes. For example, by identifying hb as a candidate gene that may have its expression distorted by the alternating temperature protocol, potentially resulting in oligomely (as discussed above), the expression of hb over time may be compared between experimental and control embryos to ascertain if the former exhibit notable deviations in expression (e.g., asymmetric expression) compared to the latter.

Modified versions of the alternating temperature protocol can also be investigated that intentionally attempt to disrupt expression of a specific gene and/or increase defect frequency; for example, by narrowing the window of treatment and exploring the application of an abrupt temperature switch at specific times relative to the height of expression for a given gene and given site(s) within embryos. This could lead to the establishment of a protocol that is able to induce certain types of anomalies with greater regularity, reducing numbers of embryos that would need to be screened for defects.

We are grateful to Daniel Rios and two anonymous reviewers for their thoughtful, insightful, and detailed comments that greatly improved the manuscript.

Additional Information and Declarations

Competing Interests

Author Contributions

Data Availability

The authors declare that they have no competing interests.

Teresa Napiórkowska conceived and designed the experiments, performed the experiments, analyzed the data, prepared figures and/or tables, authored or reviewed drafts of the article, and approved the final draft.

Julita Templin performed the experiments, prepared figures and/or tables, and approved the final draft.

Paweł Napiórkowski analyzed the data, authored or reviewed drafts of the article, and approved the final draft.

Mark A. Townley analyzed the data, authored or reviewed drafts of the article, and approved the final draft.

The following information was supplied regarding data availability:

The raw data are in the table and figures.

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
