# Peer review of "Appendage abnormalities in spiders induced by an alternating temperature protocol in the context of recent advances in molecular spider embryology"

_PeerJ, doi:10.7717/peerj.16011_

## Round 0.1 · original submission · Major Revisions

Please heed the comments of reviewers, especially of reviewers 2 and 3, before submitting a revised version of the manuscript.

Reviewer 1 ·

Basic reporting

The manuscript corresponds to a close examination and interpretation of developmental malformations due to an alternating temperature experimental manipulation on spiders. The procedure has been applied several times in previous publications, therefore the authors decide to focus in rare phenotypes not previously discussed. The authors present a rich discussion exposing the current knowledge on developmental genetics in spiders and how specific genes could correlate with their observed phenotypes. This is a very valuable addition.
Overall, I think this is an important paper to continue testing the extent of morphological variation produced under this type of environmental stress. When I say continue, I refer to the long term research program developed by this group.
The Introduction, Methods and Results are appropriate providing all the relevant information. The Discussion is well developed, but I think there are a few aspects that can be clarified or expanded. Please see below for more specific comments. My assessment is” Minor revisions”.

Experimental design

General comments:

- Line 140-141: “Our study was aimed at further exploring the diversity of developmental anomalies in postembryonic E. atrica, and this has included two rare cases of appendages on the pedicel.” The second part of the sentence (“and this has included two rare cases of appendages on the pedicel.”) does not correspond to an aim. Delete.

Validity of the findings

General comments

- Line 330-337: The addition of the genes involved in PD differentiation is very interesting and the summary presented in the Discussion appropriate. However, I think it is still missing a better connection with the phenotypes discussed. I would suggest to explicitly indicate that some of the abnormal appendages here presented do not have a clear morphological regionalization, therefore you suggest that the expression of one or more of these PD patterning genes might be altered.
- Line 459: Is there any fossil known with appendage-like structures in the pedicel? It would be useful to explicitly indicate if such a transition form exist or not.
- Discussion: Something that would be very informative is to link potential gene candidates to explain these abnormalities with the experimental procedure applied. In other words, at which point in the activation cascade of gene X (gene X corresponding to a candidate gene) could be affected by alternating temperatures? How is your experimental treatment mechanistically linked with the proposed candidate genes?
In contrast to malformations found in the wild, the authors have known environmental stressor as a cause for the malformation. Then, a further step can be taken and it could be linked with a gene which when altered produces a similar phenotype.
- Line 464-467: The paper never sets out to test if “alternating temperatures, applied during early spider embryogenesis, constitute a powerful teratogenic agent,”. Indeed, they use it as a screening tool for produce malformations. Then, this sentence should be eliminated and re-written conforming the real interest of the publication. From the Introductions the authors say: “Our study was aimed at further exploring the diversity of developmental anomalies in postembryonic E. atrica, and this has included two rare cases of appendages on the pedicel.”
- Line 469-471: “….it is possible to obtain individuals whose phenotype may reflect certain ancestral traits,”. After this sentence, add another one indicating that no fossil has been found with such a postulated transitional form (if this is the case). I.e. an arachnid with appendages in O1. It is important to not mix a speculative suggestion with a conclusion.

Additional comments

General comments

- Line 236-237: There are many mechanisms that could explain why there is a small percentage of abnormalities. Be more explicit on saying that the chaperones could be one of many mechanisms in play.
- Line 242: “…spiders must be relatively resistant to sudden temperature changes” Add a reference for this. If there is no reference change “must” by “might”
- Line 256-263: These sentences are very generic. I would summarize them in a couple of lines and immediately refer to examples of gene alterations which resemble the here presented phenotypes.
- Line 289-337: This paragraph is too long making it difficult to read. I suggest to split it in 2 or 3 paragraphs, using the start of the new paragraph the sentences where you refer in detail to the 2 confounding factors (Line 298: “Regarding (1) …) and Line 308: “Regarding (2) …”
- Line 282: I recommend the authors to double check one more time the most recent literature regarding the expression of hunchback. After a quick search, I indeed was not able to find expression in the chelicera or palps, but sometimes it is just due to the taxa surveyed.
- Line 320: “…suggesting more localized disruption to Ea-lab-1 expression.” Add the word “potential” you don’t know if this is indeed the gene affected. Then, it should read: “…suggesting a potential more localized disruption to Ea-lab-1 expression.”
- Fig. 3A: Were the protrusions labeled “a” and “b” fused with L4? As the authors do not comment on that, I imagine it is not the case. However, the insert makes it look like they are fused. For this reason, I recommend the authors to add a sentence (perhaps on the figure legend) indicating that “a” and “b” are free ends of the appendage and not fused with L4.

Minor comments:

Line 100: Change “And” by “While”
Line 141-143: “The evidence from control embryos points to all these deformities being the consequence of embryo exposure to alternating temperatures.” This sentence belongs to the Discussion.

·

Basic reporting

This work reports morphological abnormalities observed in spiders of the E. atrica species. This study reports the defects observed in the 2020/2021 season after an alternating temperature protocol, and it follows from other reports from the same group using the same alternating temperatures protocol from spiders collected in previous years (for instance see Napiórkowska et al., 2021 PeerJ). The authors report interesting phenotypes that they suggest might have important implications for the evolutionary origins of the segments of the spiders.

The text is clearly written, following the guidelines recommended by the journal.

The authors do an extensive recollection of literature in the discussion (it adds up to more than half of the paper). It would be recommended to make it more concise. While the comparison on phenotypes that emerge from knockdown studies and the alternating temperature protocol is interesting, doing direct comparisons seems far stretched. The main assumption is that temperature-shifting protocols would only affect gene expression and the whole discussion is based on that point. Nevertheless there is no evidence presented or discussed in the paper that would suggest that shifting temperature would affect the expression of specific genes (with the exception of hsp genes which is discussed in the beginning of the discussion).

As said, the discussion is interesting but might be out of scope for this paper and perhaps would be more interesting for a review.

Lines 233-235: "we obtained 212 individuals with body defects in the 2020/2021 breeding season, representing 7% of the successfully hatched postembryos, and about 4% of the embryos, in the experimental group." - it seems that the percentages are inverted.

Experimental design

The authors mention that in control conditions they found that 10% of eggs did not hatch, whereas in the temperature-shifting protocol this number was 40%. Is it possible that this 10% corresponds to non-fertilized eggs? They briefly mention that they randomly checked that the embryos were fertilized by immersing in paraffin but they do not mention how many did they inspect.

The authors have published this temperature-shifting protocol earlier. It would be interesting to mention how similar the results are from different collecting seasons - whether they see similar percentages of lethality and abnormalities, and whether they see novel phenotypes they have not found earlier.

Validity of the findings

As mentioned earlier, the authors discuss how loss of different genes could explain the phenotypes that they found on their samples, but they do not consider other explanations. Given that their data is only based on the description of phenotypes, they should at least discuss other possibilities, for instance, the role of metabolism in development and how this could be affected by temperature, or mechanical defects and how these could be linked to temperature shifts, like cell migration, cell division, growth etc., without necessarily affecting gene expression.

The authors should make a clear reference to their previous works and how this one adds up to what they have found in previous breeding seasons

Reviewer 3 ·

Basic reporting

The authors present the results of a study of developmental defects in spiders caused by applying alternating temperatures. This study is part of a decades-long study of the impact of temperature on the development of spiders. The authors present some unique phenotypes that were recovered in their recent experiments. They also present data regarding an interesting phenotype in which appendages are found on the first opisthosomal segment, which normally lacks appendages. The data are of high quality and the manuscript is very well written. A weakness of the manuscript is that the discussion is highly speculative by attempting to assign the defects recovered with disruption of the function of specific genes. Additionally, it is unclear what benefit there is in speculating about the precise cause of these defects.

Minor Comments

Line 94: I recommend deleting “first”.

Table 1: Based on the title of Table 1, It is not clear whether these numbers are coming from all specimens analyzed (control + experimental) or only experimental specimens. I understand that the control didn’t show any abnormalities based on the text in the results section, but many readers will just skim through the data. Clarify in the Table title that this data is for the experimental group only and clarify in the title that no defects were seen in the control group. Also, change “others” to “other”, as in “other abnormalities”.

Major Comments:

In general, it is unclear what is gained by speculating about the developmental genetic cause of the phenotypes that are described. Additionally, the genes of focus are primarily transcription factors. Therefore, even if the defects described are related to a candidate transcription factor, they are not necessarily caused by loss of expression of the candidate gene or misexpression of the candidate gene. The ultimate cause could be disruption of the function of a gene or genes that regulate the candidate gene, or disruption of the function of any downstream target of the candidate gene.

Discussion: Oligomeric abnormalities: Are these specimens missing evidence of segmentation beyond a missing appendage? At least some of them are missing the entire hemisegment according to the authors. Should specimens that are only missing an appendage be scored separately from those that are missing an entire hemisegment? Additionally, even with Dll RNAi, the proximal parts of appendages are still present. Therefore, the loss of appendages in the experiments group may be better explained as a partial loss of a segment. Do you notice any lateral to medial bias in terms of loss of structures, which could help explain the range of defects seen in the oligomeric abnormalities?

Lines 309-311: Why wait for a future study to utilize the number and arrangement of slit organs to identify the leg segments that are defective? If this is possible, it should be done for this study.

Lines 328-337: It is unclear why disruption of exd/hth function is hypothesized to cause the loss of chelicerae and pedipalp. The function of these genes has not been established in spiders. In harvestmen, interfering with hth function leads to homeotic transformations of these appendages to leg, but not loss. In insects, appendages are lost when entire head segments are deleted in RNAi targeting these genes. In no case does RNAi targeting either hth or exd lead specifically to a loss of an appendage without the loss or reduction of the corresponding segment.

Lines 394-421: The first abdominal segment of insects also exhibits a Distal-less expressing appendage, the pleuropodia, during embryogenesis. This similarity with spiders should be considered in decisions of whether an embryonic appendage constitutes a redefinition of a tagma.

Experimental design

No comment.

Validity of the findings

No comment.

---

## Round 0.2 · Minor Revisions

Please heed the comments from reviewer #1 so that the article is acceptable for publication.

Reviewer 1 ·

Basic reporting

I have carefully review the authors’ replies to the reviewers’ comments and re-assessed the whole manuscript. I can see an overall improvement and an increase in the precision of the statements. But, I still think that an important aspect to make clear across the text is that the authors are discussing potential developmental explanations to the observed phenotypes and not proposing them as a unique explanation. In general this message comes across, but I would encourage the authors to re-read all the statements one more time to make sure that it is the case all throughout. Their experimental treatment has decades of publications demonstrating how unspecific it is in terms of producing malformations (the same treatment can produce a plethora of malformations). Therefore, more than in any other experimental treatment, I would suspect that the here presented phenotypes are the result of multiple developmental alterations.
Other than this important overarching clarification, I think the manuscript is a valuable contribution which brings up new data, propose future experiments and interesting developmental and evolutionary ideas.

All the evaluation criteria positively achieved.

Experimental design

- The aims read a bit confusing. I would suggest to summarize them in something like: “Our study was aimed to document and discuss potential developmental explanations to rare (ie. appendages on the pedicel) anomalies in postembryonic E. atrica induced by the alternating temperature protocol.” As for the last two sentences, starting with “Note that aberrant…”, I would suggest to move this explanation to the previous paragraph. Its presence in the paragraph with the aims of the study just makes it less clear overall.

All the other evaluation criteria positively achieved.

Validity of the findings

- At the end of the newly added section of “Summary and future directions”, the authors suggest that modifications in the alternating temperature protocol could allow to “more consistently generate certain defects would increase the feasibility of monitoring a gene’s expression over time”. Although, I agree with the statement and fine tuning the method could allow to replicate some type of malformations more frequently. I don’t think that is the most efficient approach to achieve the goal of “monitoring a gene’s expression over time”. Techniques like RNAi are significantly more precise and reliable. The wide variety of phenotypes obtained from the alternating temperature protocol shows how unspecific it is. Then, I would recommend to tone down this perspective and present the alternating temperature protocol mainly as a screening tool.
- The temporal changes on the mortality rate and % of hatching individuals is quite striking. As a reader I appreciate this overview. I saw that you do not have an explanation, but I would strongly encourage the authors to try to look further into this. Maybe there are changes in the experimental procedure, or the way to inspect individuals; or perhaps, there has been a change on the natural populations? Has there been a change in temperature (median, lowest low, highest high, etc.) in the study area during the last decade?
It could be interesting to add as a future direction the further study on this change. It might be particularly interesting in the current context of climate change.
- Do the authors have any explanation as to why oligomely was the most frequent anomaly? It would be important to present those ideas at the end of the paragraph between lines 255-281.
- Line 296: I would not cite the cases of Pt-foxQ2, Pt-six3.1, or Pt-six3.2, Pt-Delta and Pt-Sox21b to explain the lethality of the experimental treatment. There are just too many possibilities to explain that outcome, then referring to these specific genes could be misleading.
- I agree with the authors in the value of looking in the literature for potential candidate genes to explain the observed phenotypes in the temperature alternating experiments. But, in some sentences, it appears that given the morphological similarity the authors directly propose that the responsible genes was found. I would strongly recommend to revise the wording use on those types of sentences.
See Line 470-472. An improvement here would be to indicate: “the potential alteration in could affect Ea-Antp-1 or any of the up or downstream genes associated, and their respective developmental processes.”

All the other evaluation criteria positively achieved.

Additional comments

- Reviewer 2 points out that the authors always refer to changes in gene expression, while other developmental processes could be affected. The authors reply indicating that they will use a more generic term including those changes, the term selected was “aberrant gene expression”. In my opinion it is the same as before. The focus still on gene expression. I do understand the author’s argument in saying: “even if levels of expression of a given gene remain normal, the distribution of gene products will likely be positionally altered from normal”. But, I think a better generic term would be: “developmental alterations” or “developmental anomalies”.

Specific comments:

- Line 135: add comma after the word “latter”
- Line 274: “prosomata” Is this correct?
- Line 293: add comma after the word “segments”
- Line 298: Review the redaction in the sentence: “Also lethal and relevant is embryonic development that,”. It reads very confusing.
- Line 383: add comma after the word “pedipalps”
- Line 386: add comma after parenthesis.
- Line 402: At the end of the sentence add: “or other developmental processes.”
- Line 433: Change “suggested” by “hypothesized”
- Line 457: add comma after the word “legs”

·

Basic reporting

The authors have addressed the comments suggested in the previous round of revisions. Specifically, they mention that the developmental defects observed in their protocol could emerge not only from changes in gene expression but also due to changes in metabolism or other cellular processes that could be affected by temperature fluctuations. They include an interesting comparison between this and previous harvests. They also explain that the literature revision they include in the discussion will be useful for planning future experiments, and that these may help understand the phenotypes they have identified so far.

Experimental design

No further comments

Validity of the findings

No further comments

Reviewer 3 ·

Basic reporting

No comment.

Experimental design

No comment.

Validity of the findings

No comment.

Additional comments

The authors have responded to all of my comments. I have no further comments. Overall, they present an interesting study that moves toward an approach to connect environmentally induced defects during spider embryogenesis to known developmental regulatory mechanisms.

---

## Round 0.3 · Minor Revisions

Please heed comments by the reviewer.

Reviewer 1 ·

Basic reporting

- Line 136-139: I suggest to simplify the following sentence: “We also sought to consider abnormalities like those seen in the 2020/2021 breeding season, which included two rare cases of appendages on the pedicel, in terms of potential errors in developmental gene expression.”. I would recommend to just say: “In particular, we further discussed the potential developmental mechanisms of two rare appendages on the pedicel.”

Experimental design

No comments.

Validity of the findings

As the authors will see, all the following comments represent very minor and localized improvement suggestions to the manuscript. In the majority of the cases asking to tone down a statement or move a paragraph.

- Line 267: In the revised version the authors have added a potential explanation to the increase on the mortality rate (but, see my comments below). This addition would be better placed after the phenomenon is discussed in the text and not isolated at the end of the manuscript. I suggest to move the paragraph from Lines 560-571 to the beginning of the discussion (after Line 267).
On the added text the authors do not present a summary or a future direction, instead they present a potential explanation for an unexpected observation. That is why I recommend the movement.
- Line 569: I would recommend to revise the proposed hypothesis for the increase in mortality rate. The authors argue that the return of control individuals to their collection site might create inbreeding depression. However, if the collection for the experiment would have not been carried out, the same individuals would potentially reproduce on site. Perhaps, the fact that some individuals from the population were removed due to the experimental treatment could justify a smaller population with consequently higher inbreeding. However, it does not appear as a strong argument.
I suggest to add a sentence explaining this situation, or propose another hypothesis.
- Line 271: After presenting the fact that oligomelies are the more common malformation it would be more appropriate to include the recently added text with the potential explanation for this observation. I am referring to the text currently corresponding to the Lines 572-581. Currently, it seems out of place to end the paper with this very specific comment, which is not even at the center of the discussion.
- Line 301-304: I would recommend to tone down or even delete the last sentence of the paragraph. It is hard to conclude anything from embryos without evident malformations, which are not able to hatch. There are many alternatives to explain that phenomenon and associated with three specific genes might be misleading.
- Line 307: “…and congruent with application of our thermal treatment…” How are those genes congruent with the experimental treatment? It would be better to say something like “genes whose knock down have a similar phenotype to the one produced by our thermal treatment”.
- Line 335: I recommend to tone down this sentence. As currently written, it suggest a potential inhibition of two specific genes due to the experimental protocol. Although, it is possible, it is also possible that other genes or developmental processes create that phenotype. Moreover, it appears very unlikely that such non-specific treatment (thermal oscillation) would act more specifically than an RNAi procedure.
- Line 365: "Thus, thermally-induced defects consistent with inhibited Ea-Dll expression might actually reflect initial direct disruptions to Ea-Sp6-9 expression." This is a big assumption. Thermally-induced defects consistent with inhibited Ea-Dll expression could actually reflect any disruptions on the pathway in which Dll is implicated, as well as any other gene in appendage development. I recommend to tone down that sentences under the consideration of my comment.
The fact that a known gene produces a similar phenotype as a broad effect experimental manipulation, it does not mean that the experimental manipulation is unequivocally affecting that specific gene.
- Line 405: I would not use as an argument for a localized perturbation the fact that the thermal oscillation experiments yield asymmetric phenotypes. Instead, I would argue that it shows how given the determinate type of development of most of protostome animals, alterations on one side of the organism, do not necessarily affect the other. Just that.
- Line 553: The paragraph starting with: “Modified versions of the alternating temperature protocol can also be…” would be a better paragraph to close the paper than the ones added on this last version. I recommend to move it to the end.
- Regarding my previous comment on pin pointing specific genes as potential causes of lethality, I acknowledge the efforts of the authors on being cautious on their statements, as well as their interest on providing the evidence that alterations on such genes have resulted in lethal consequences. However, I still think that it would be important to explicitly add a sentence indicating that there are many other mechanisms to explain the lethal outcome.
For a more specific phenotypic result (ie. a teratology), it does make sense to invoke a potential candidate gene. However, something so generic such as the dead of an organism after an experimental treatment of very broad spectrum, it appears to be less necessary. In common words, there are just too many ways to die.

Additional comments

- Line 250-254: I would delete the two last sentences of this paragraph, because they do not contribute anything new. As it is presented in the Introduction, the method of alternating temperatures has been used for decades to generate developmental abnormalities. Therefore, I do not see the point to argue that it “appears” to be “effective”.

---

## Round 0.4 · accepted · Accept

Congratulations! You paper has been accepted for publication in PeerJ.

Reviewer 1 ·

Basic reporting

I have reviewed the authors responses and their respective implementations in the main text. I do not have additional comments. My assessment is: Accepted.
Congratulations for their work and all the improvements during the review process.

Experimental design

No comments.

Validity of the findings

No comments.

Additional comments

No comments.